# Design and antiviral assessment of a panel of fusion proteins targeting human papillomavirus type 16

Chongzhi Bai [1,2,3]☯*, Ruoyu Wang[1]☯, Qian Yang[1], Jianqing Hao[4], Qiming Zhong[1], Ruiwen Fan[3], Pengcheng Han[5]*

1 Central Laboratory, Shanxi Province Hospital of Traditional Chinese Medicine, Taiyuan, China, 2 CAS Key Laboratory of Pathogenic Microbiology and Immunology, Institute of Microbiology, Chinese Academy of Sciences (CAS), Beijing, China, 3 College of Veterinary Medicine, Shanxi Agricultural University, Taigu, Jinzhong, China, 4 School of Basic Medical Sciences, Shanxi Medical University, Taiyuan, China, 5 School of Medicine, Zhongda Hospital, Southeast University, Nanjing, China

☯ These authors contributed equally to this work.
* baicz@im.ac.cn (CB); 101013216@seu.edu.cn (PH)

## Abstract

Cervical cancer ranks as the third most prevalent malignancy in women worldwide. The persistence of Human papillomavirus (HPV) infection stands out as the foremost risk factor for cervical cancer development. Among the numerous HPV subtypes, HPV16 infection emerges as the primary pathogenic determinant of cervical cancer. To date, no specific drugs have been approved. In this study, we engineered two high-affinity fusion protein targeting HPV16 L1 protein based on the alpaca-derived single-domain antibody 2C12 previously obtained in our laboratory. These two fusion proteins exhibited potent neutralizing activity against HPV16 pseudovirus with IC50 values of 7.8 nM and 6.5 nM, respectively. Molecular docking analysis revealed that 2C12 formed ten pairs of hydrogen bonds with HPV16 L1, among which Arg39 and Thr100 established multiple pairs of hydrogen bonds with HPV16 L1, indicating their crucial roles in antigen-antibody binding process. These structural and biological findings underscore the effective binding capacity of these fusion proteins to HPV16, leading to reduced viral load and providing valuable insights into therapeutic antibody and vaccine development against HPV 16 infection.

## Introduction

HPV is a non-enveloped, double-stranded circular DNA virus that primarily infects basal keratinocytes on the surface of the skin and mucosa [1]. Cervical cancer represents a significant cause of mortality among women worldwide, with approximately 600,000 new cases and over 310,000 female deaths reported annually [2, 3]. Over the past decade, there has been a substantial increase in cervical cancer incidence and an alarming trend towards younger age at onset, making it one of the deadliest diseases with rapidly growing mortality rates. Nearly all cases of cervical cancer are caused by infection with high-risk HPV [4, 5]. Among more than 100 HPV types identified so far, around 14 have been associated with carcinogenesis [6]. HPV16 is recognized as a high-risk subtype responsible for over half of all cervical cancers and precancerous

**Funding:** This study was financially supported by Shanxi Provincial Key Research and Development Project in the form of a grant (2022ZDYF094) received by CB. This study was also financially supported by Basic Research Project of Shanxi Province in the form of a grant (202303021212353) received by RW. This study was also financially supported by National Outstanding Youth Science Fund Project of National Natural Science Foundation of China in the form of a grant (32222006) received by PH.

**Competing interests:** The authors have declared that no competing interests exist.

lesions [7]. Without intervention measures in place, approximately 50% of patients diagnosed with grade 3 cervical intraepithelial neoplasia face an elevated risk for developing cervical cancer due to persistent atypical hyperplasia of skin lesions [8]. The current treatment primarily involves cryotherapy, laser ablation, loop electrosurgical excision procedure and adjuvant therapy, such as interferon, which fails to address the issue of subclinical and latent infection, resulting in symptomatic relief rather than addressing the underlying cause and increasing the likelihood of relapse.

The major capsid protein L1 is the main component of the icosahedral capsid of HPV, with a molecular weight of 50–60 kD. It is the main target of immune killing against HPV virus, the expression of L1 is closely related to the replication of HPV virus and the early stage of infection [9]. The L1 proteincan be expressed alone and self-assembled into virus-like particles (VLPs), induces HPV-type-specific neutralizing antibodies [10, 11]. The structure of VLPs is highly similar to that of the natural virus, which is the core of the research and development of HPV preventive vaccine. The human body can be immunized with VLPs to stimulate the production of high titer neutralizing antibodies [12]. Through the injection of prophylactic HPV vaccine, the human body is immunized to activate the immune system in the body to produce specific antibodies and memory lymphocytes secreting such antibodies to prevent subsequent HPV infection [13], it is an effective active immunization method. However, prophylactic HPV vaccines do little to eradicate preexisting infection, because the L1 capsid protein is not expressed on the surface of infected basal epithelial cells [14]. Therefore, prophylactic HPV vaccine is not an effective treatment for people already infected with HPV [9, 15]. In addition, through passive immunization, a large number of traditional antibodies are prepared in vitro, which can quickly neutralize and encapsulate HPV when used in the skin and mucosa layer to prevent HPV infection. Although this mode can quickly neutralize the virus in vitro, due to the poor pH/temperature stability of the traditional antibody, it cannot form a lasting prevention mechanism on the mucosal surface-.Specifically, a decrease in the number of antibodies on the mucosal surface presents an opportunity for HPV virus to infiltrate and induce viral infection [16].

Alpaca -derived single-domain antibodies, also referred to as nanoantibodies (Nbs), are small peptides (approximately 15 kD) that bind to antigens. These Nbs contain three extended complementarity-determining regions (CDRs) which allow them to effectively target epitopes that are typically inaccessible to conventional antibodies, such as viral domains hidden by glycans [17–19]. In contrast to traditional antibodies, Nbs can be produced at a reduced cost, have high solubility, and can be generated in large quantities through microbial engineering techniques [20, 21]. Neutralizing antibodies, Nbs, and combination strategies have been employed for the prevention of immune evasion [22–25]. Moreover, it is relatively easy to bioengineer Nbs in order to enhance their neutralizing activity while simultaneously reducing manufacturing expenses and scale requirements [20, 26]. Given these aforementioned advantages over traditional antibodies, Nbs hold great potential for the development of drugs against pathogenic microorganisms.

In this study, we utilized the alpaca-derived single-domain antibody 2C12 to construct two fusion proteins capable of binding HPV16 L1 with high affinity and effectively neutralizing HPV16 pseudovirus. Through structure prediction and molecular docking, we elucidated the potential molecular mechanism and identified key binding sites of 2C12 in its interaction with HPV16 L1.

## Materials and methods

### Source of materials

Protein Ladder was purchased from Thermo Fisher SCIENTIFIC (26616). 293FT cell line was purchased from Nanjing Vazyme Biotechnology Co. LTD. BL21 (DE3) and pColdII vector

**Table 1. AHPVD and AHPVT fusion protein expression plasmids.**

| Fusion protein | Nanobody | Plasmid backbone | Resistance Marker |
|---|---|---|---|
| AHPVD | bivalent 2C12, 15aa length GS linker | PcoldII | Ampicillin |
| AHPVT | trivalent 2C12, 15aa length GS linker | PcoldII | Ampicillin |

was purchased from GENEWIZ Biotechnology Co. LTD. Expression plasmids pCMVHPV16-L1-flag-L2 and reporter plasmids pcDNA3.1-EGFP-HPV16-E6/E7 were synthesized by General Biology Systems Co., LTD. HPV16 L1 protein, alpaca-derived single domain antibody 2C12 and Z6 (Anti-ZIKV human mAb Z6) were prepared and stored in our laboratory [27]. Isopropyl-β-D-thiogalactoside was purchased from Sangon Biotechnology Co. LTD.

## Construction, expression and purification of fusion protein

The previously obtained sequence of laboratory-derived single-domain antibody protein 2C12 was subjected to codon optimization, and the resulting peptide was linked with $(GGGGS)_3$ protein to generate a dimer (designated as AHPVD) and a trimer fusion protein (designated as AHPVT). The fusion protein sequence, containing an N-terminal His tag, was cloned into the pColdII vector (Table 1). Subsequently, the constructed vector was transformed into BL21 (DE3) competent cells and these cells were inoculated in LB medium. Upon reaching an $OD_{600}$ value of 0.6, induction of expression was achieved by adding 0.5 mM isopropyl-β-D-thiogalactoside at 15˚C overnight. -Centrifugation at 4000×g for 10 min at 4˚C followed by high-pressure crushing machine treatment, the lysate was further centrifuged at 12000×g for 40 min at 4˚C to collect the supernatant. 2C12, AHPVD and AHPVT proteins were purified via affinity chromatography using a HisTrap HP 5 mL column (Cytiva) and the target proteins were eluted in an elution buffer composed of 20 mM Tris (pH 8.0), 150 mM NaCl, and 300 mM imidazole. The samples were then purified using gel-filtration chromatography on a HiLoad 16/600 Superdex 75 prep grade gel filtration chromatography column (Cytiva) in a buffer containing 20 mM Tris (pH 8.0) and 150 mM NaCl, and with the flow rate of 1ml/min.

## Identification of the affinity of fusion proteins to HPV16 L1

HPV16 L1 protein was immobilized on Chip CM5 (Cytiva) chip, and the binding kinetics of fusion proteins to HPV16 L1 were detected by surface plasmon resonance (SPR) instrument Biacore T200 (GE Healthcare). The buffer system was PBST (10 mM $Na_2HPO_4$, 2 mM $KH_2PO_4$, 137 mM NaCl, 2.7 mM KCl, 0.005% Tween 20, pH 7.4). A mock-coupled surface was used for the background subtraction. IC50

After each assay, the cells were regenerated with 10 mM glycine (pH 3.0) for 2 min. Binding data were fitted with the Langmuir binding equation using a 1:1 interaction model by Biacore™T200 evaluation software. These results were then visualized using Graphpad Prism 8.

## HPV16 pseudovirus construction

The structural gene expression plasmids and reporter plasmids were routinely transformed, extracted in large quantities, and subsequently stored at -20˚C for future use. 293FT cells were seeded at a density of $1×10^6$/mL in 6-well plates and cultured overnight at 37˚C. Transfection experiments were conducted when the cell density reached approximately 70%. A single reporter plasmid was transfected as a negative control group. For transfection experiments, 293FT cells were seeded at a density of $1.5×10^4$/mL in 96-well plates and cultured overnight at 37˚C in a $CO_2$ incubator (Thermo) with a concentration of 5%. HPV16 pseudovirus was

diluted by factors of 0.1, 0.01, and then further diluted to concentrations of $1\times10^{-3}$, $1\times10^{-4}$, and finally to reach dilutions of up to $1\times10^{-5}$ times using DMEM medium. Each gradient dilution (100 µL) was added to multiple wells (8 wells per dilution gradient) containing the seeded 293FT cells. The positive wells were identified as those exhibiting more than one diseased cell under an inverted fluorescence microscope. Cell culture infection was quantified using the Reed-Muench method based on half quantity (half of tissue culture infective dose TCID50). The TCID50 end point represents the virus dilution degree where half the virus infects the host cells with a dilution ratio of approximately 1:1000.

## Pseudovirus neutralization experiments

Transfected $1\times10^4$ 293FT-HPV16 L1 cells were spread in 96-well plates and grown to 90% at 24 h, and the marginal Wells of the plates were filled with 200 µL PBS to prevent evaporation. Fusion proteins were serially diluted 3-fold in DMEM with 10% fetal bovine serum, ranging from 1µM to 0.46 nM. The diluted pseudovirus (1:1000 dilution) was added into a 96-well plate, diluted fusion proteins were added, distilled water was used as the control group, and the cells were incubated at 37˚C for 1 h. 100 µl of the above mixture was added to the 293FT cell Wells. After incubation at 37˚C for 24 h, green fluorescence (reporter gene production) was detected by CQ1 confocal quantitative image cytometer (Yokogawa) microscopic reading method, photographed, and counted. Each group contained at least three replicates. Percent neutralization was normalized by considering cells infected without fusion proteins as 0% neutralization and uninfected cells as 100% neutralization. The half-maximal neutralizing concentration (IC50) was calculated by four-parameter nonlinear regression analysis using GraphPad Prism 8 software.

## Stability analysis

The thermostability test protein of 2C12 and fusion protein was diluted to a concentration of 1 mg/mL using a Tris-Cl buffer (0.05 mol/L) and subjected to incubation at various temperatures including 4˚C, 25˚C, 37˚C, 49˚C, and 61˚C for a duration of 15 m. Subsequently, the protein samples were rapidly cooled in an ice water bath. Indirect ELISA was performed to determine the titer under different dilutions (1:50, 1:200, 1:800, 1:1600, 1:3200, 1:6400 and 1:12800), with each condition having six replicates. The negative control group Z6 was included.

The acid-base stability test proteins of 2C12 and fusion proteins were diluted to 1 mg/mL with 0.05 mol/L Tris-Cl buffer (pH 3–9), and then immediately neutralized with 2 mol/L Tris solution (pH 7.0) at 37˚C for 4 h. At different dilutions of 1:50, 1:200, 1:800, 1:1 600, 1:3 200, 1:6 400 and 1:12 800, each condition was set up 6 complex wells, and the titer was detected by indirect ELISA. The negative control group Z6 was included.

## Protein structure acquisition and molecular docking

Based on the amino acid sequences, alphafold2 was used to predict the 3D structures of the two fusion proteins, and AlphaFold selected the top ranked optimized structures by classifying the models according to the average pLDDT was 91.805. Based on the pLDDT of the protein, the render was performed using UCSF Chimera 1.16. Further validation and quality assessment of the 3D structure were performed, the ERRAT quality score was 97.0588, the Ramachandran plot: 94.6% core, 5.4% allow, 0.0% gener, 0.0% disall, Z-Score: -6.08 (S1 and S2 Figs). Molecular docking of the 2C12 to HPV16 L1 was performed using ZDock 3.0.2 software, which uses full-space search sampling and can filter out the closest native conformation from 2000 predicted structures. The docking process combines the factors of spatial structure,

electrostatic interaction and conformational change of the fusion protein with HPV16 L1. HPV16 L1 with known structure (PDB: 3j6r) was selected as the receptor protein, and the fusion protein with predicted structure was used as the ligand protein for docking and calculation. The top 10 predictions of the docking output file were selected, which contained 2000 docking simulation conformations sorted by ZDock Score, and the model with the highest score was selected for analysis.

## Results

### Expression and purification of fusion proteins AHPVD and AHPVT

The AHPVD and AHPVT proteins were purified to high purity using nickel ion column affinity chromatography and gel filtration chromatography. Based on the comparison with the HiLoad 16/600 Superdex 75 prep grade protein standard curve, AHPVD and AHPVT eluted at about 70 min (or 70 mL) and 60 min (or 60 mL), respectively, and the curve are symmetrical (Fig 1A and 1B). The single target protein bands corresponding to their theoretical molecular weights of 30 kD and 45 kD were observed on a 12.5% SDS-PAGE gel, and were stained with coomassie brilliant blue G250 (Fig 1C). (Amount of protein loading: marker 5 µL, AHPVD 5 µg, AHPVT 10 µg). Combined with gel filtration chromatography and SDS, we can infer that the obtained proteins were relatively homogeneous and highly purified.

### Enhanced binding activity of the AHPVD and AHPVT against the HPV16 L1

Surface plasmon resonance (SPR) was employed to investigate the binding kinetics between fusion protein and HPV16 L1. The affinities of AHPVD and AHPVT towards HPV16 L1 were determined as 5.11 nM and 3.26 nM, respectively, both exhibiting higher affinity compared to single-domain antibody 2C12 with an affinity of 30.12 nM towards HPV16 L1 (Fig 2 and S1 Table). To accurately assess the neutralization potency of the two fusion proteins against HPV16 L1 protein, a pseudovirus-luciferase reporter assay was conducted. Consistent with the affinity assay results, at the same concentration of 9.45 nM, AHPVD and AHPVT demonstrated significantly improved neutralizing activity against HPV16 pseudovirus when compared to single-domain antibody 2C12 (Fig 3A–3D), with IC50 values of 7.8 nM for AHPVD and 80 nM for 2C12 respectively (Fig 3E and S2 Table). Notably, AHPVT exhibited even higher neutralization activity with an IC50 value of 6.5 nM suggesting its superior binding ability in efficiently targeting multiple HPV16 virus particles.

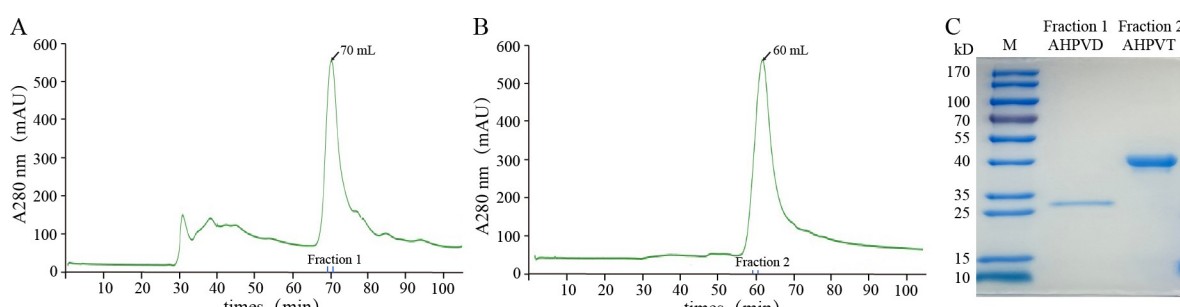

**Fig 1. Purification and identification of AHPVD and AHPVT protein.** (A) Results of AHPVD protein HiLoad 16/600 Superdex 75 prep grade gel filtration chromatography. (B) Results of AHPVT protein HiLoad 16/600 Superdex 75 prep grade gel filtration chromatography. (C) SDS-PAGE identification of AHPVD and AHPVT protein.

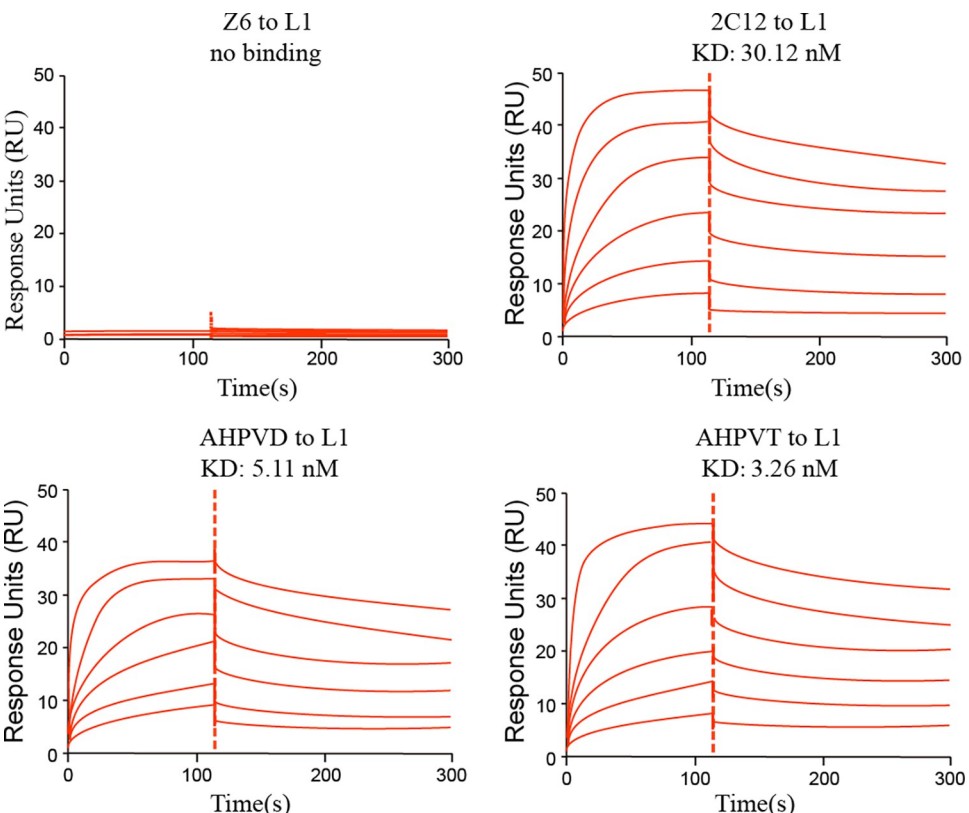

**Fig 2. 2C12, AHPVD and AHPVT binding affinities to HPV 16 L1.** Z6 was used as a negative control. The red color represents the fitted curve. The binding profiles are shown with time (s) on the x-axis and response units (RU) on the y-axis. KD represents the equilibrium dissociation constant, KD values shown are the mean ± standard deviation (SD) of three independent experiments.

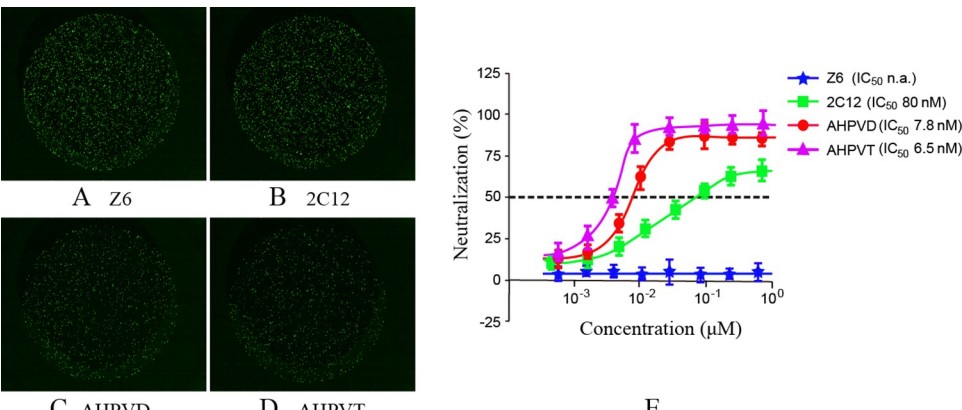

**Fig 3. The neutralizing capacity of 2C12, AHPVD and AHPVT for HPV 16 pseudoviruses: Z6 was used as a negative control.** (**A**) Z6; (**B**) 2C12; (**C**) AHPVD; (**D**) AHPVT; (**E**) The green fluorescencewas detected by CQ1 microscopic reading method. Three independent experiments were performed with two replicates. The curves and IC50 values are one representative data, in which the error bar for each concentration is presented as mean ± SD.

Table 2. Thermal stability.

| Temperature(°C) | 2C12 (titer) | AHPVD (titer) | AHPVT (titer) |
|---|---|---|---|
| 4 | 1:6400 | 1:6400 | 1:6400 |
| 25 | 1:6400 | 1:6400 | 1:6400 |
| 37 | 1:6400 | 1:6400 | 1:6400 |
| 49 | 1:6400 | 1:6400 | 1:6400 |
| 61 | 1:1600 | 1:1600 | 1:1600 |

## Stability study

The thermal stability test revealed no change in the titer of the fusion protein following a 15-minute water bath at various temperatures. However, when the temperature reached 61°C, there was a significant reduction in antibody titer, as presented in Table 2. The acid-base stability test depicted in Table 3 demonstrated that the fusion protein's antibody titer remained unchanged during a 4-hour water bath at pH levels ranging from 4 to 8. Nevertheless, substantial reduction in antibody titer occurred at pH values of both 3 and 9.

## Structural basis of fusion proteins binding to HPV16 L1

The 3D structure of 2C12 was predicted using alphafold2, while molecular docking between 2C12 and the known HPV16 L1 (PDB:3j6r) structure was performed using ZDock3.0.2. The results revealed that 2C12 formed ten pairs of hydrogen bonds with HPV16 L1 (Fig 4A). Specifically, R39 formed three pairs of hydrogen bonds with I117 and Y49 of HPV16 L1, S112 formed one pair of hydrogen bonds with Y355 of HPV16 L1, and T100 formed two pairs of hydrogen bonds with R258 of HPV16 L1. Additionally, it established a hydrogen bond interaction with Y418 and H259 as well as S99 and N101 respectively, involving V257 and E130 residues in HPV16 L1. Notably, R39 and T100 in 2C12 exhibited multiple strong hydrogen bonding interactions with HPV16 L1, indicating their crucial roles in antigen-antibody binding process (Fig 4B–4D). Furthermore, extensive hydrophobic interactions were observed between the two proteins resulting in a significant surface contact area; the total surface area between them measured approximately 1095:1105 (Fig 4E), which allows the two proteins to bind tightly.

## Discussion

Therapeutic antibody drugs have rapidly emerged as a significant component of modern biomedicine in recent years. Cervical cancer, the third most prevalent malignancy in women, is primarily caused by high-risk HPV infection. Therefore, the development of antibody designs capable of binding multiple epitopes and exhibiting high blocking activity holds great

Table 3. Acid-base stability.

| pH | 2C12 (titer) | AHPVD (titer) | AHPVT (titer) |
|---|---|---|---|
| 3 | 1:1600 | 1:1600 | 1:800 |
| 4 | 1:6400 | 1:6400 | 1:6400 |
| 5 | 1:6400 | 1:6400 | 1:6400 |
| 6 | 1:6400 | 1:6400 | 1:6400 |
| 7 | 1:6400 | 1:6400 | 1:6400 |
| 8 | 1:6400 | 1:6400 | 1:6400 |
| 9 | 1:800 | 1:800 | 1:800 |

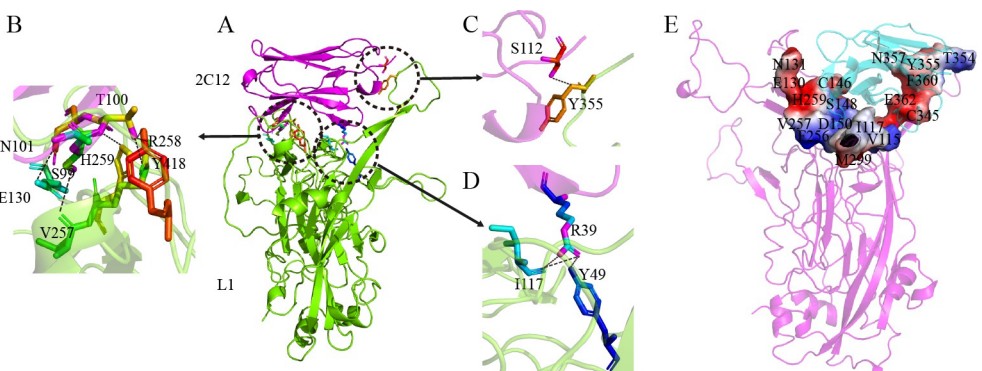

**Fig 4. Structure analysis of 2C12 complex with the HPV16 L1. (A)** Ribbon representations of the complex of 2C12 (pink) with the HPV16 L1 (green), the hydrogen region is highlighted by black circle in the side view; **(B-D)** Zoomed-in view of 2C12 with HPV16 L1 polar interaction; **(E)** Interaction surface is displayed in surface representation model and colored in red and blue, key residues are labeled.

importance in the absence of specific drugs. Such designs can provide effective candidate antibody drugs and targets for preventing and treating HPV16 infection. Nano-antibodies derived from camels possess only the variable region of heavy chain with a molecular weight ranging from 12 to 15 kD, which is one-tenth that of traditional antibodies [28]. These nano-antibodies exhibit complete antigen-binding ability and are superior to ordinary antibodies in neutralizing viruses [29]. Additionally, they offer advantages such as easy expression, modification, high stability, and tissue permeability [30]. We identified a panel of Omicron BA.1 spike receptor-binding domain (RBD)-targeted nanobodies (Nbs) from a naive alpaca VHH library [28]. This panel of Nbs exhibited high binding affinity to the spike RBD of wild-type. Through multivalent Nb construction, we obtained several fusion proteins with strong neutralization against a variety of SARS-CoV-2 mutant pseudoviruses [28]. The potential of these fusion proteins is promising and may serve as novel drug candidates for the treatment of viral infections.

Professor Broom isolated specific nanobodies against the HPV-16 major capsid protein L1, which are capable of specific recognition of the L1 epitopes and have low cross-reaction with irrelevant antigens [31]. This research proved that anti-HPV-16 L1 nanobodies are highly desirable molecules for the future development of novel diagnostic and therapeutic methods against human papillomavirus. In this study, we modified and constructed alpaca-derived single-domain antibody 2C12 against HPV16 into a recombinant plasmid suitable for prokaryotic expression. After modification and successful purification, we obtained dimer fusion protein (AHPVD) and trimer fusion protein (AHPVT). Compared to the single-domain antibody 2C12 alone, AHPVD and AHPVT exhibited approximately 6–10 times higher binding affinity towards HPV16 L1. Furthermore, pseudovirus neutralization assay demonstrated their enhanced neutralizing activity compared to 2C12 alone. It is speculated that due to having more binding sites for HPV16 L1 on AHPVD and AHPVT molecules themselves efficiently bind multiple HPV16 virus particles resulting in stronger binding ability leading to higher neutralizing activity against pseudovirus. The stability study of the single-domain antibody and the two fusion proteins demonstrated that all three proteins maintained high antibody titers even after incubation at 49˚C. Furthermore, there was no significant change in antibody titers across a pH range of 4 to 8, indicating excellent temperature and pH stability. We attempted to determine the crystal structure of the complex formed by 2C12 and HPV16 L1, however, we were unable to obtain suitable crystals for screening. Molecular docking analysis using ZDock3.0.2 revealed that 2C12 and HPV16 L1 formed five pairs of hydrogen bonds as

well as multiple hydrophobic interactions, resulting in a total surface area ratio of 1095:1105 between HPV16 L1 and 2C12. Hydrogen bonding and hydrophobic interactions are the most important interactions between protein molecules, and these interaction forces enable the formation of interaction interfaces between proteins. The kinetic properties or binding properties of proteins will be altered when proteins form tight binding, leading to changes in its function. Notably, Thr100 of 2C12 exhibited three pairs of hydrogen bonds with HPV16 L1 while also forming numerous contact surfaces around this residue. Conserved presence of Thr100 in other HPV16 L1 llama-derived single-chain antibodies (scAbs) sequences (unpublished data) suggests its importance as a binding site for anti-HPV16 L1 antibodies. Docking results further indicated that Arg258 on HPV16 L1 engaged in two pairs of hydrogen bonds with Thr100 on 2C12, highlighting strong interaction between these sites and suggesting Arg258 as another crucial binding site for potential antibody-based therapies against HPV16.

Prophylactic HPV vaccines are primarily protein-based VLPs, they can trigger both humoral and cellular immune responses [32, 33], and have been approved for the prevention of cervical cancer in populations without preexisting HPV infection. Therapeutic HPV vaccines primarily target E6 and E7 proteins are also in clinical tests [34]. However, to date, no therapeutic HPV vaccine has been approved. The nanoantibodies obtained in this study present good pH/ temperature stability. They can effectively reduce the viral load, and may form a long-lasting therapeutic mechanism on the surface of cervical mucosa. They show great advantages in the application of preventing and treating cervical cancer caused by persistent HPV infection.

Given the advantages of nanoantibodies, more options for the prevention and treatment of HPV infection will be provided. However, the current nano-protein preparations for HPV prevention and control are still far away for human use, with uncertainty concerning. Their safety, stability, potency and rapid production still need to be further studied.

## Supporting information

**S1 Table. 2C12, AHPVD and AHPVT binding affinities to HPV 16 L1.**
(XLSX)

**S2 Table. The neutralizing capacity of 2C12, AHPVD and AHPVT for HPV 16 pseudo-viruses: Z6 was used as a negative control.**
(XLSX)

**S1 Fig. Ramachandran plot.**
(JPG)

**S2 Fig. ERRAT quality score.**
(JPG)

## Acknowledgments

We thank Prof. George Fu Gao and Prof. Qihui Wang for their kind support in language refinement and manuscript revision.

## Author Contributions

**Conceptualization:** Chongzhi Bai, Ruiwen Fan, Pengcheng Han.

**Data curation:** Ruoyu Wang, Jianqing Hao.

**Formal analysis:** Jianqing Hao.

**Investigation:** Ruoyu Wang.

**Methodology:** Qian Yang.

**Project administration:** Qian Yang.

**Resources:** Qian Yang, Qiming Zhong.

**Validation:** Qiming Zhong.

**Writing – original draft:** Ruoyu Wang, Qian Yang.

**Writing – review & editing:** Chongzhi Bai, Jianqing Hao, Qiming Zhong, Ruiwen Fan, Pengcheng Han.

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
