## [Decision Letter · Decision Letter 0]

5 Aug 2024

PONE-D-24-26930Design and antiviral assessment of a panel of fusion proteins targeting human papillomavirus type 16PLOS ONE

Dear Dr. Bai,

Thank you for submitting your manuscript to PLOS ONE. After careful consideration, we feel that it has merit but does not fully meet PLOS ONE’s publication criteria as it currently stands. Therefore, we invite you to submit a revised version of the manuscript that addresses the points raised during the review process.

We look forward to receiving your revised manuscript.

Kind regards,

Haitham Mohamed Amer, PhD

Academic Editor

PLOS ONE

Reviewers' comments:

Reviewer's Responses to Questions

**Comments to the Author**

1. Is the manuscript technically sound, and do the data support the conclusions?

Reviewer #1: Partly

Reviewer #2: Yes

2. Has the statistical analysis been performed appropriately and rigorously? 

Reviewer #1: I Don't Know

Reviewer #2: I Don't Know

3. Have the authors made all data underlying the findings in their manuscript fully available?

Reviewer #1: No

Reviewer #2: Yes

4. Is the manuscript presented in an intelligible fashion and written in standard English?

Reviewer #1: Yes

Reviewer #2: Yes

5. Review Comments to the Author

**Reviewer #1:** Overview: This study aimed to engineer, express, purify, and characterize two immunogenic fusion proteins targeting HPV 16-L1 protein. The study further aims to compare it to 2C12, a nanobody previously generated by these investigators against the same target. Computational analysis was performed to characterize the residues critical to target binding further. Given that only prophylactic vaccines, and not therapeutic vaccines, are available for HPV, the work addresses an important need and the use of nanobodies seems to be a novel and scalable approach.

Introduction The authors do a nice job of providing an introduction to nanobodies and summarizing their key findings.

Weaknesses - For clarity and relevance to the current study, it would have been helpful to make a stronger case as to why a therapeutic vaccine would be important in the field and why current prophylactic vaccines are insufficient to address the public health issue of HPV-mediated cancers.

An excellent review on this topic can be found at: https://doi.org/10.3389/fimmu.2024.1362770

Methods

Can the authors provide additional detail? At present, there is insufficient detail to replicate the work in this report. The manufacturers of the reagents and analyzers used to collect data are not provided. For example, the authors used pColdII as the parent vector to create the fusion protein construct. Where did they obtain pColdII? For the IMAC purification, which resin did they use, how did they elute the Ni-column – no details or data are provided. The specific MW standards used were not described for the gel filtration chromatography and SDS PAGE.

For the Biosensor experiments, the authors do not provide details of their assay, such as the concentration range and specific concentration of the fusion proteins for each analysis. They also don’t describe performing blank runs. Was this done?

For the pseudovirus inhibition experiments, can the authors please provide more details on what the CQI microscopy reading method entails, including a reference? Also, how were the images obtained? Line 127 has a typo.

Results

Purification – the authors seem to have purified the fusion proteins to near homogeneity; however, the lack of specific details makes this hard to determine conclusively. The yield of the fusion proteins was not described.

Figure 1. The gel filtration standards should be named, and their elution volumes should be indicated on the chromatogram. SDS PAGE across the gel filtration fractions should be provided to demonstrate that they correspond to AHPVP and AHPVT. What are the commercial names and providers of the MW standards used in the gel filtration and SDS-PAGE?

Figure 1C. What percentage of gel is shown? How much protein is run in each lane. How are the bands detected?

Figure 2. The blank run is not provided. I could not find the analyte concentrations used in each run on the sensograms. This information should be provided. What exactly is “Z6”? Please provide the numerical data that corresponds to the sensograms.

Figure 3. The resolution and low signal make the images hard to interpret. Based on the curve shown in Figure 3E, there should be a significant difference between the images shown in Figure 3A and 3B – as shown, they seem identical. Also, the curves shown for AHPVD and AHPVT don’t seem to reflect the differences in the signal shown in the images.

Figure 4. What templates were used to create the models in Alphafold2? Two different abbreviation conventions are used to name the amino acids in this one figure: the three-letter code and the one-letter code. The authors should stick to one approach. The average pLDDT values were not provided.

Discussion

The authors didn’t really discuss their work in the context of previous work done – they simply reiterated their results.

**Reviewer #2: **1. Justify the role of major capsid protein L1 with elaborative explanation and proper references.

2. The language and sentence making of methods section need to be polished. Few sentences are not clearly expressing the performed steps.

3. Why alphafold2 server was used, there are many other validated servers for structure prediction including Itasser, Trosetta etc. Justify the use of this specific server in this study?

4. The 3D structure of 2C12 was predicted using alphafold2, but further validation and quality assessment of the 3D structure is missing. A proper validation which includes ERRAT quality score, favored region in Ramachandran plot and z score need to be provided with figure.

5. Some of the explanation in discussion needs proper explanation. For example, this line “Molecular docking analysis using ZDock3.0.2 revealed that 2C12 and HPV16 L1 formed five pairs of hydrogen bonds as well as multiple hydrophobic interactions”, what is the significance of hydrogen bond and hydrophobic interaction. Appropriate justification with reference is vital.

6. Compare some of the findings with previous study which is missing in the discussion. Is there any similar study or findings exist for AHPVD and AHPVT, compare with other relevant studies?

7. Illustrate figure 4 more clearly, you can improve the quality and resolution of the picture.

6. PLOS authors have the option to publish the peer review history of their article (what does this mean?). If published, this will include your full peer review and any attached files.

Reviewer #1: No

Reviewer #2: No

---

## [Author Response · Author response to Decision Letter 0]

28 Aug 2024

Response to Reviewers

Comments to the Author

1. Is the manuscript technically sound, and do the data support the conclusions?

Reviewer #1: Partly

Reviewer #2: Yes

2. Has the statistical analysis been performed appropriately and rigorously?

Reviewer #1: I Don't Know

Reviewer #2: I Don't Know

3. Have the authors made all data underlying the findings in their manuscript fully available?

Reviewer #1: No

Reviewer #2: Yes

4. Is the manuscript presented in an intelligible fashion and written in standard English?

Reviewer #1: Yes

Reviewer #2: Yes

5. Review Comments to the Author

Reviewer #1: Overview: This study aimed to engineer, express, purify, and characterize two immunogenic fusion proteins targeting HPV 16-L1 protein. The study further aims to compare it to 2C12, a nanobody previously generated by these investigators against the same target. Computational analysis was performed to characterize the residues critical to target binding further. Given that only prophylactic vaccines, and not therapeutic vaccines, are available for HPV, the work addresses an important need and the use of nanobodies seems to be a novel and scalable approach.

Introduction The authors do a nice job of providing an introduction to nanobodies and summarizing their key findings.

Weaknesses - For clarity and relevance to the current study, it would have been helpful to make a stronger case as to why a therapeutic vaccine would be important in the field and why current prophylactic vaccines are insufficient to address the public health issue of HPV-mediated cancers.

An excellent review on this topic can be found at: https://doi.org/10.3389/fimmu.2024.1362770

Response: Thank you for the comments and the literature sharing. We really appreciate your acknowledgement of our studies and also see your concerns in many aspects. As suggested, we have carefully read the literature and added the description that why a therapeutic vaccine would be important in the field and why current prophylactic vaccines are insufficient to address the public health issue of HPV-mediated cancers. (line 61-65)

Methods

Can the authors provide additional detail? At present, there is insufficient detail to replicate the work in this report. The manufacturers of the reagents and analyzers used to collect data are not provided. For example, the authors used pColdII as the parent vector to create the fusion protein construct. Where did they obtain pColdII? For the IMAC purification, which resin did they use, how did they elute the Ni-column – no details or data are provided. The specific MW standards used were not described for the gel filtration chromatography and SDS PAGE.

Response: Thanks for the suggestion. We have provided the manufacturers of the reagents and analyzers used to collect data in the revised manuscript. (line 92-100 , line112-118 and line 205)

For the Biosensor experiments, the authors do not provide details of their assay, such as the concentration range and specific concentration of the fusion proteins for each analysis. They also don’t describe performing blank runs. Was this done?

Response: Thank you for your kind suggestion. We have provided details of the assay in the revised manuscript. (line 124-129)

For the pseudovirus inhibition experiments, can the authors please provide more details on what the CQI microscopy reading method entails, including a reference? Also, how were the images obtained? Line 127 has a typo.

Response: Thank you for your kind suggestion. We have provided details of the assay and revised the typo in the revised manuscript. (line 160-162)

Results

Purification – the authors seem to have purified the fusion proteins to near homogeneity; however, the lack of specific details makes this hard to determine conclusively. The yield of the fusion proteins was not described.

Response: Thank you for your kind suggestion. Indeed, it seems hasty to draw this conclusion. We have revised the sentence as follows “Based on the comparison with the HiLoad 16/600 Superdex 75 prep grade protein standard curve, AHPVD and AHPVT eluted at about 70 min and 60 min, respectively, and he curve is symmetrical (Fig 1A and 1B). The single target protein bands corresponding to their theoretical molecular weights of 30 kD and 45 kD were observed on a 12.5% SDS-PAGE gel (Fig 1C). Combined with gel filtration chromatography and SDS-PAGE gel, we can infer that relatively homogeneous and highly purified proteins are obtained.” (line 200-208)

Figure 1. The gel filtration standards should be named, and their elution volumes should be indicated on the chromatogram. SDS PAGE across the gel filtration fractions should be provided to demonstrate that they correspond to AHPVP and AHPVT. What are the commercial names and providers of the MW standards used in the gel filtration and SDS-PAGE?

Response: Thank you for your kind suggestion. As you suggested, we have provided the details about the gel filtration chromatography in the “Materials and methods” part (line112-118), revised the figure legend and resubmitted Figure 1. The commercial names and providers of the MW standards used in the gel filtration and SDS-PAGE have been provided. (line93)

Figure 1C. What percentage of gel is shown? How much protein is run in each lane. How are the bands detected?

Response: Sorry, we didn't describe it clearly in the previous manuscript. We have revised this section. (line203-207)

Figure 2. The blank run is not provided. I could not find the analyte concentrations used in each run on the sensograms. This information should be provided. What exactly is “Z6”? Please provide the numerical data that corresponds to the sensograms.

Response: Thank you for your kind suggestion. As you suggested, We have provided details of the assay in the revised manuscript. (line 126-129)

Z6 was used as a negative control, it is Anti-ZIKV human mAb, which was prepared and stored in our laboratory. (line 97-99)

Figure 3. The resolution and low signal make the images hard to interpret. Based on the curve shown in Figure 3E, there should be a significant difference between the images shown in Figure 3A and 3B – as shown, they seem identical. Also, the curves shown for AHPVD and AHPVT don’t seem to reflect the differences in the signal shown in the images.

Response: I'm sorry to have caused you trouble. In fact, the resolution of figure 3 meets the requirements. It may be due to the distortion of the resolution after uploads, which makes it impossible to identify. Please download the TIF format image to view. 

As described in the manuscript line 219-221 “AHPVT exhibited even higher neutralization activity with an IC50 value of 6.5 nM suggesting its superior binding ability in efficiently targeting multiple HPV16 virus particles”. In order to more clearly demonstrate the higher neutralizing activity of AHPVT and AHPVD than 2C12, we selected the pictures of the four proteins at 6.5 nM concentration Figure 3A-3D. The neutralization activity of four proteins at this concentration were approximately Z6 was no more than 5%, 2C12 was no more than 10%, AHPVD was 30%, AHPVT was 50%, respectively. From the images, it is difficult to obtain a clear difference between Figure 3A and 3B, also between Figure 3C and 3D. We have uploaded the data in the Supporting information.

Figure 4. What templates were used to create the models in Alphafold2? Two different abbreviation conventions are used to name the amino acids in this one figure: the three-letter code and the one-letter code. The authors should stick to one approach. The average pLDDT values were not provided.

Response: Thank you for your kind suggestion and sorry for the trouble brought to you in the review. We predicted the structure using default Settings and did not use templates. We have revised the Figure 4 as you suggested, and the average pLDDT values were 91.805(line 178-180).

Discussion

The authors didn’t really discuss their work in the context of previous work done – they simply reiterated their results.

Response: Thank you for your kind suggestion. We have revised the manuscript.

Reviewer #2: 

1. Justify the role of major capsid protein L1 with elaborative explanation and proper references.

Response: Thank you for your kind suggestion. We have revised the manuscript and added some descriptions of the role of major capsid protein L1. (line 51-53 and line62-63)

2. The language and sentence making of methods section need to be polished. Few sentences are not clearly expressing the performed steps.

Response: Thank you for your kind suggestion. We have revised the methods section.

3. Why alphafold2 server was used, there are many other validated servers for structure prediction including Itasser, Trosetta etc. Justify the use of this specific server in this study?

Response: Thank you for your kind suggestion. There are 113 amino acids in 2C12, its structure is not complicated, and we are familiar with alphafold2, which was also used in our previous study for protein structure prediction (26). After the structure prediction, further validation and quality assessment of the 3D structure were performed, and the predicted results were reasonable.

4. The 3D structure of 2C12 was predicted using alphafold2, but further validation and quality assessment of the 3D structure is missing. A proper validation which includes ERRAT quality score, favored region in Ramachandran plot and z score need to be provided with figure.

Response: Thank you for your kind suggestion. We have provided description about validation and quality assessment of the 3D structure in the revised manuscript. (Line182-184) Further validation and quality assessment of the 3D structure were performed, the ERRAT quality score was 97.0588, the Ramachandran plot: 94.6% core, 5.4% allow, 0.0% gener, 0.0% disall, Z-Score: -6.08 (Supporting Information:Figure S1 and S2). The Z-score is plotted by the online version of the tool ProSa, which is not loaded.

5. Some of the explanation in discussion needs proper explanation. For example, this line “Molecular docking analysis using ZDock3.0.2 revealed that 2C12 and HPV16 L1 formed five pairs of hydrogen bonds as well as multiple hydrophobic interactions”, what is the significance of hydrogen bond and hydrophobic interaction. Appropriate justification with reference is vital.

Response: Thank you for your kind suggestion. We have added the description about hydrogen bonds and multiple hydrophobic interactions in the revised manuscript. (line237-250 and line 293-297) 

6. Compare some of the findings with previous study which is missing in the discussion. Is there any similar study or findings exist for AHPVD and AHPVT, compare with other relevant studies?

Response: Thank you for your kind suggestion. We have added the discussion of previous studies (line263-269 and line270-274). 

7. Illustrate figure 4 more clearly, you can improve the quality and resolution of the picture.

Response: Thank you for your kind suggestion. We have revised the manuscript and Figure 4.

---

## [Decision Letter · Decision Letter 1]

10 Sep 2024

PONE-D-24-26930R1Design and antiviral assessment of a panel of fusion proteins targeting human papillomavirus type 16PLOS ONE

Dear Dr. Bai,

Thank you for submitting your manuscript to PLOS ONE. After careful consideration, we feel that it has merit but does not fully meet PLOS ONE’s publication criteria as it currently stands. Therefore, we invite you to submit a revised version of the manuscript that addresses the points raised during the review process.

We look forward to receiving your revised manuscript.

Kind regards,

Haitham Mohamed Amer, PhD

Academic Editor

PLOS ONE

Journal Requirements:

Reviewers' comments:

Reviewer's Responses to Questions

**Comments to the Author**

1. If the authors have adequately addressed your comments raised in a previous round of review and you feel that this manuscript is now acceptable for publication, you may indicate that here to bypass the “Comments to the Author” section, enter your conflict of interest statement in the “Confidential to Editor” section, and submit your "Accept" recommendation.

Reviewer #1: All comments have been addressed

Reviewer #2: All comments have been addressed

2. Is the manuscript technically sound, and do the data support the conclusions?

Reviewer #1: Yes

Reviewer #2: Yes

3. Has the statistical analysis been performed appropriately and rigorously? 

Reviewer #1: Yes

Reviewer #2: N/A

4. Have the authors made all data underlying the findings in their manuscript fully available?

Reviewer #1: Yes

Reviewer #2: Yes

5. Is the manuscript presented in an intelligible fashion and written in standard English?

Reviewer #1: Yes

Reviewer #2: Yes

6. Review Comments to the Author

Reviewer #1: (No Response)

Reviewer #2: (No Response)

7. PLOS authors have the option to publish the peer review history of their article (what does this mean?). If published, this will include your full peer review and any attached files.

Reviewer #1: No

Reviewer #2: No

---

## [Author Response · Author response to Decision Letter 1]

11 Sep 2024

1. Line 48 – the term” physical therapy” has a very specific meaning, and their usage in this context is incorrect. The authors should state what clinical intervention for atypical hyperplasia/neoplasia they are referencing, such as Cryotherapy, Laser ablation, Loop electrosurgical excision procedure (LEEP). ...Cone biopsy, etc.

Response: Thank you for your kind suggestion. We have revised this section. (line 46-49)

Response: Thank you for your kind reminder. We have revised the reference 19 correctly. (line 382-384)

---

## [Decision Letter · Decision Letter 2]

16 Sep 2024

Design and antiviral assessment of a panel of fusion proteins targeting human papillomavirus type 16

PONE-D-24-26930R2

Dear Dr. Bai, 

We’re pleased to inform you that your manuscript has been judged scientifically suitable for publication and will be formally accepted for publication once it meets all outstanding technical requirements.

Kind regards,

Haitham Mohamed Amer, PhD

Academic Editor

PLOS ONE

---

## [Editor Report · Acceptance letter]

15 Oct 2024

PONE-D-24-26930R2 

PLOS ONE

Dear Dr. Bai, 

I'm pleased to inform you that your manuscript has been deemed suitable for publication in PLOS ONE. Congratulations! Your manuscript is now being handed over to our production team.

Kind regards, 

on behalf of

Dr. Haitham Mohamed Amer 

Academic Editor

PLOS ONE